# Control of spin-wave transmission by a programmable domain wall

Sampo J. Hämäläinen[1], Marco Madami [2], Huajun Qin[1], Gianluca Gubbiotti [3] & Sebastiaan van Dijken [1]

Active manipulation of spin waves is essential for the development of magnon-based technologies. Here, we demonstrate programmable spin-wave filtering by resetting the spin structure of pinned 90° Néel domain walls in a continuous CoFeB film with abrupt rotations of uniaxial magnetic anisotropy. Using micro-focused Brillouin light scattering and micromagnetic simulations, we show that broad 90° head-to-head or tail-to-tail magnetic domain walls are transparent to spin waves over a broad frequency range. In contrast, magnetic switching to a 90° head-to-tail configuration produces much narrower and strongly reflecting domain walls at the same pinning locations. Based on these results, we propose a magnetic spin-wave valve with two parallel domain walls. Switching the spin-wave valve from an open to a closed state changes the transmission of spin waves from nearly 100 to 0%. Active control over spin-wave transport through programmable domain walls could be utilized in magnonic logic devices or non-volatile memory elements.

[1] NanoSpin, Department of Applied Physics, Aalto University School of Science, P.O. Box 15100, FI-00076 Aalto, Finland. [2] Dipartimento di Fisica e Geologia, Università di Perugia, 06123 Perugia, Italy. [3] Istituto Officina dei Materiali del CNR (CNR-IOM), Sede Secondaria di Perugia, c/o Dipartimento di Fisica e Geologia, Università di Perugia, 06123 Perugia, Italy. Correspondence and requests for materials should be addressed to S.v.D. (email: sebastiaan.van.dijken@aalto.fi)

Wave-like computing based on spin waves has generated interest as a potential low-power and parallel computing alternative for conventional CMOS technologies[1]. High group velocities[2], long decay lengths[3], and the ability to tailor the wavelength of spin waves down to the nanoscale[4–6] offer fascinating prospects for magnonics. Logic devices based on a Mach–Zehnder spin-wave interferometer have been proposed and realized[7–10]. In this geometry, spin waves are launched into the interferometer branches, and their phase or amplitude is controlled by an Oersted field[7–9] or spin-wave current from a third magnetic terminal[10]. Interference of spin-wave signals after manipulation determines the logic output. Other logic concepts exploit reprogrammable magnonic crystals[11,12]. Here, spin-wave transmission is controlled actively by a spatial modulation of magnetic properties using, for instance, electric currents[13] or optical pulses[14]. Besides these dynamic approaches, field-induced toggling between ferromagnetic and antiferromagnetic states in magnetic stripe arrays has been shown to modify spin-wave transport in a non-volatile way[15,16].

Non-collinear spin structures, such as magnetic domain walls, can also be exploited to control the amplitude or phase of spin waves[17,18]. It has been analytically calculated that spin-wave transport through an infinitely extended one-dimensional Bloch wall induces a finite phase shift without reflection[19]. In contrast, interactions between spin waves and 180° Néel walls or magnetic domain walls in confined geometries are more complex[20]. Dynamic stray fields in such domain walls reduce the transmission of spin waves if their wavelength exceeds the wall width[21,22]. Additionally, domain-wall resonances limit the transmission of spin waves at specific frequencies[23–26]. This effect, known as resonant reflection, relates strongly to the spatial inhomogeneity of the effective magnetic field inside the wall. Thus, because of dynamic stray fields and resonance modes, narrow domain walls reflect spin waves more than broad walls. In many studies, the interaction between spin waves and magnetic domain walls is investigated as a new method to drive walls into motion[22–29]. It has been shown that the direction and velocity of wall motion depend strongly on the spin-wave transmission coefficient. For coefficients close to unity, transfer of angular momentum causes domain walls to move against the spin waves[27,28]. In contrast, strong spin-wave reflection induces forward domain-wall motion[22–26].

In this paper, we propose the use of programmable domain walls for active spin-wave manipulation (Fig. 1a). Periodic 90° rotations of uniaxial magnetic anisotropy firmly pin Néel-type walls in our system, preventing their motion under the action of propagating spin waves. Because of pinning, a magnetic field can reversibly transform a narrow 90° head-to-tail domain wall into a broad 90° head-to-head or tail-to-tail wall. Using phase-resolved micro-focused Brillouin light scattering (μ-BLS) and micromagnetic simulations, we show that broad domain walls are transparent for spin waves over a wide frequency range. In contrast, spin waves are resonantly reflected by narrow domain walls.

## Results

**Experimental realization.** Interactions of spin-waves with 90° Néel walls have been imaged in square-shaped ferromagnetic microstructures with a Landau domain state[30,31]. The head-to-tail domain walls in these experiments act as barriers for spin-wave transport. To realize active switching between such narrow domain walls and broader 90° walls with a head-to-head or tail-to-tail structure, we couple a 50 nm thick ferromagnetic CoFeB film to a ferroelectric BaTiO₃ substrate with regular ferroelastic stripe domains (Fig. 1b). The combination of strain transfer and inverse magnetostriction in this bilayer system causes imprinting of magnetic stripe domains in the ferromagnetic layer[32,33]. The magnetic anisotropy of the CoFeB film is uniaxial, reflecting the tetragonal symmetry of the underlying ferroelectric crystal, and the anisotropy axis rotates abruptly by 90° between domains. In a previous study, we showed that excitation of this system by an uniform microwave magnetic field results in the formation of standing spin waves within the domains and spin-wave emission from the anisotropy boundaries[34]. Here, we focus on spin-wave transport through 90° Néel walls. Importantly, strong pinning of straight magnetic domain walls at the anisotropy boundaries enables switching between head-to-tail and head-to-head/tail-to-tail walls in the CoFeB film. We use an external in-plane magnetic field to toggle between these non-volatile magnetization states. If the field is applied along the domain boundaries and turned off, the magnetization aligns in an alternating head-to-head and tail-to-tail configuration (Fig. 1c, top). The same process after rotating the in-plane field by 90° initializes head-to-tail domain walls

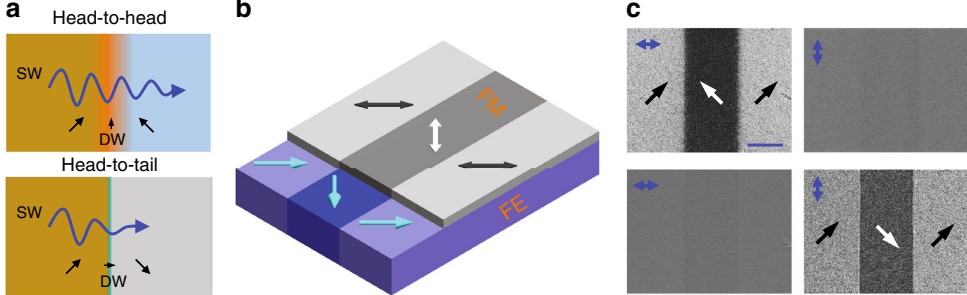

**Fig. 1** Spin-wave control via programming of pinned magnetic domain walls. **a** Schematic of spin-wave (SW) transmission through a pinned 90° magnetic domain wall (DW) with a head-to-head and head-to-tail structure. The pinned domain wall in the ferromagnetic film separates stripe domains with uniaxial magnetic anisotropy. Because the anisotropy axis rotates by 90° at the domain boundary, a magnetic field along the boundary initializes a head-to-head (top) or tail-to-tail wall. These domain walls are broad and transparent to spin waves. Application of a magnetic field perpendicular to the boundary results in the formation of a much narrower head-to-tail domain wall (bottom). Spin waves are mostly reflected by this wall. Reversible magnetic switching of the domain-wall structure thus actively alters the transmission of spin waves. **b** Illustration of the experimental CoFeB/BaTiO₃ bilayer system. The domain pattern of the ferroelectric (FE) substrate is imprinted into the ferromagnetic (FM) film via strain transfer and inverse magnetostriction. Blue arrows and double-headed black and white arrows indicate the direction of ferroelectric polarization and the orientation of uniaxial magnetic anisotropy. **c** Magneto-optical Kerr microscopy images of the magnetization configuration in a CoFeB/BaTiO₃ bilayer after the application of a magnetic field along the domain boundaries (top) and perpendicular to the domain boundaries (bottom). The left and right images are taken with the magneto-optical contrast axis along the horizontal and vertical direction, respectively (blue arrows). The blue scale bar corresponds to 10 μm

(Fig. 1c, bottom). Both walls are of the 90° Néel type in zero magnetic field, but their widths differ significantly[35]. Scanning electron microscopy with polarization analysis (SEMPA), X-ray photoemission electron microscopy (XPEEM), and micro-magnetic simulations on CoFeB/BaTiO₃ and related systems have previously shown that the width ($\delta$) of 90° head-to-head/tail-to-tail domain walls increases sharply with CoFeB film thickness ($t$) because of their large magnetostatic energy ($\delta \propto t$)[36–38]. In contrast, the width of 90° head-to-tail domain walls is almost entirely determined by a competition between ferromagnetic exchange and the strength of magnetic anisotropy and, thus, varies much less with $t$[36]. Consequently, the difference in domain wall width is more pronounced in thick ferromagnetic films. In our experiments, we focus on 50-nm-thick CoFeB because it combines full imprinting of magnetic stripe domains (Fig. 1b, c) with the ability to reprogram the domain wall width from ~50 nm (head-to-tail wall) to 1.6 μm (head-to-head and tail-to-tail wall). Details on sample preparation are given in the Methods section.

**Brillouin light scattering experiments.** Dispersion relations of spin waves depend on the angle between the wave vector and direction of magnetization. In most spin-wave experiments, a magnetic bias field is used to saturate the magnetization of a sample either parallel or perpendicular to the wave vector. Because of strain-induced uniaxial magnetic anisotropy in the CoFeB film of our bilayer, the magnetization is uniform within the stripe domains in zero magnetic field. This enabled us to perform measurements without bias field and to study spin-wave transport through 90° Néel walls. In the remanent state, the CoFeB magnetization aligns along the uniaxial magnetic anisotropy axes of the stripe domains, i.e., at an angle of 45° with respect to the domain walls (Fig. 1c). The type of magnetic domain wall was set before spin-wave characterization by either applying a parallel or perpendicular magnetic field.

We employed μ-BLS to measure spin-wave transmission through pinned magnetic domain walls. For the excitation of spin waves, we patterned 500-nm-wide microwave antennas on top of our sample using electron-beam lithography. The antennas are separated from the CoFeB film by an insulating TaO$_x$ layer, and they are aligned parallel to a nearby domain wall (Fig. 2a, top). The microwave antennas excite spin waves over a broad

range of wave vectors (the shortest wavelength is about 1 μm). In the experiments described below, the antenna edge and domain-wall center are separated by ~2 μm (Fig. 2a, bottom).

Figure 2b shows μ-BLS spectra measured at a fixed location 4.5 μm from the antenna edge (i.e., 2.5 μm beyond the center of the pinned magnetic domain wall) and by scanning the rf excitation frequency in the range 5–15 GHz. The black and red curves depict BLS intensity data for a broad 90° head-to-head and narrow 90° head-to-tail domain wall, respectively. Two main spin-wave resonances are measured. The peaks measured at ±12.6 GHz are the same for both domain walls. We attribute these resonances to thermally excited perpendicular standing spin waves (PSSWs) in the CoFeB film because the peaks are recorded also when the rf-source is turned off and the dependence of their frequency on external magnetic field closely follows the calculated dispersion of the first-order PSSW mode (Supplementary Fig. 1). The other main resonances ranging from ±7.7 to ±12.2 GHz in the μ-BLS spectra of Fig. 2b illustrate the intensity of propagating spin waves after transmission through the magnetic domain wall. Since we maintained all experimental conditions, the BLS intensities for both magnetization configurations can be compared directly. The local μ-BLS measurements indicate that spin waves are transmitted through the broad 90° head-to-head domain wall and that their intensity is significantly suppressed by the narrow 90° head-to-tail domain wall. The same conclusions can be drawn from phase-resolved μ-BLS scans across the magnetic domain walls (Fig. 2c). In these phase-resolved measurements, μ-BLS line scans were performed along the orange arrow in Fig. 2a. While the signal intensity drops significantly behind the head-to-tail domain wall, we do not measure a similar suppression after we switch the magnetization to the head-to-head configuration.

Figure 3 shows μ-BLS line scans (a, d) and 2D intensity scans (b, c, e, f), measured at two different excitation frequencies. The measurements illustrate the decay of spin-wave intensity from the antenna edge across the pinned domain wall at 9.85 GHz (a–c) and 10.85 GHz (d–f). Line scans recorded at two other frequencies are shown in Supplementary Fig. 2. The data corroborate that initialization of a narrow 90° head-to-tail domain wall at $x = 0$ μm causes strong spin-wave reflection (red curves in Fig. 3a, d and 2D intensity scans in Fig. 3b, e). The BLS intensity after the pinned head-to-tail domain wall is almost

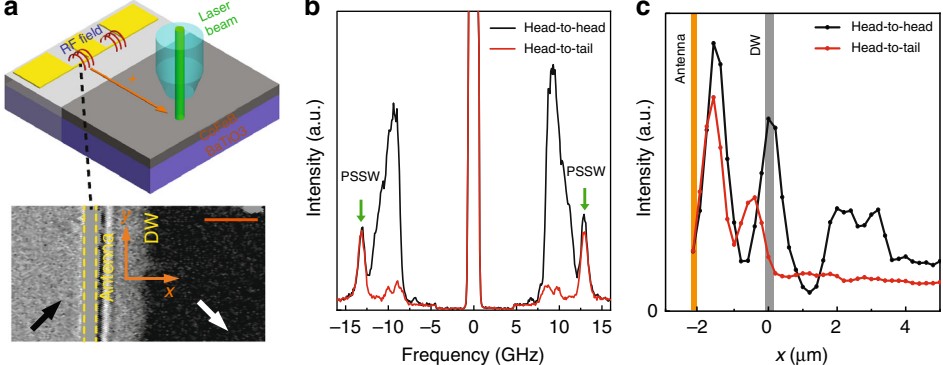

**Fig. 2** Brillouin light scattering of spin-wave transmission. **a** Schematic and magneto-optical Kerr microscopy image of the μ-BLS measurement geometry. A microwave antenna on top of the CoFeB film excites propagating spin waves. Transmission of these spin waves through a nearby head-to-head or head-to-tail domain wall is recorded by scanning the μ-BLS laser beam across the wall (orange arrow in top panel). The orange scale bar in the bottom panel corresponds to 2 μm. **b** μ-BLS spectra measured at a fixed position, 4.5 μm from the antenna, and by scanning the rf excitation frequency in the range 5–15 GHz. Dissimilar transmission of propagating spin waves through the broad 90° head-to-head and narrow 90° head-to-tail domain wall is illustrated by the intensity peaks that range from ±7.7 to ±12.2 GHz. The peak recorded at 12.6 GHz on both the Stokes and anti-Stokes side of the spectra indicates thermal excitation of a PSSW mode in the CoFeB film. **c** Phase-resolved μ-BLS scans across the pinned domain wall for head-to-head and head-to-tail magnetization configurations. The excitation frequency in this measurement is 10.85 GHz

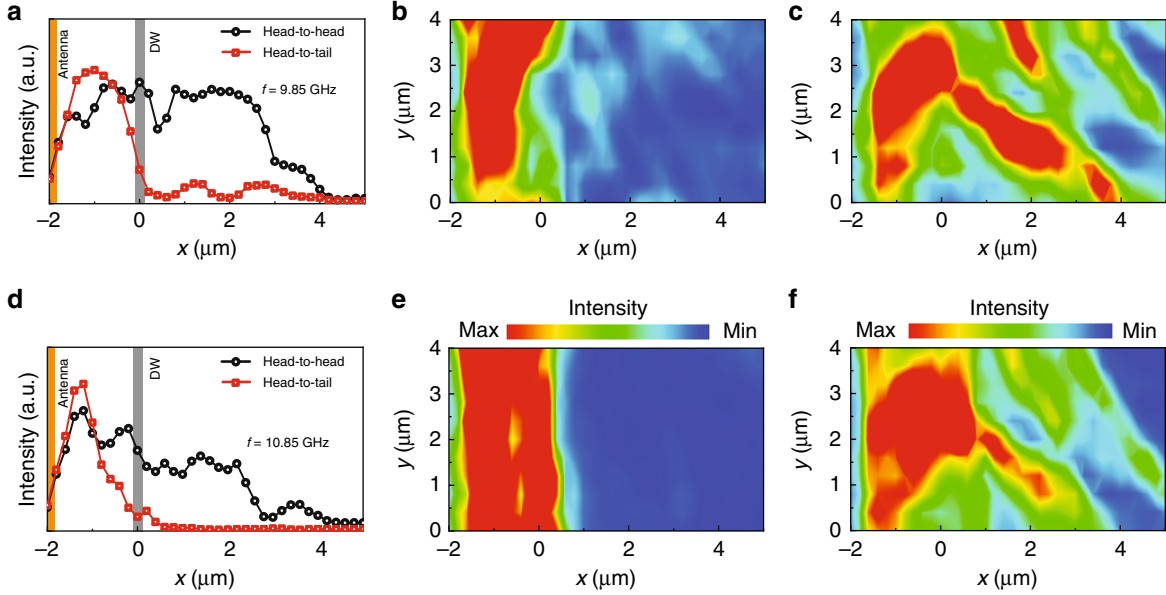

**Fig. 3** Frequency and spatial dependence of spin-wave transmission. Line scans and areal maps of BLS intensity recorded from the antenna edge across the pinned domain wall at excitation frequencies of 9.85 GHz (**a–c**) and 10.85 GHz (**d–f**). The areal maps in **b**, **e** are recorded after the magnetization configuration is set to a head-to-tail domain wall. The maps in **c**, **f** show BLS intensity data for a head-to-head domain wall

completely suppressed to the noise level at 10.85 GHz, whereas a small signal is measured at 9.85 GHz. In this magnetization configuration, the spin-wave signal already decays before the pinned domain wall. We attribute this effect to destructive interference of forward moving and reflected spin waves. The suppression is most pronounced at 10.85 GHz, providing additional proof that spin-wave reflection is particularly strong at this frequency. Switching the magnetization of the CoFeB film to a broad 90° head-to-head domain wall drastically enhances the transmission of spin waves at all frequencies (black curves in Fig. 3a, d and 2D intensity scans in Fig. 3c, f). In the 2D scans, we observe spatial non-uniformities in the spin-wave intensity. Similar BLS-intensity variations are also measured if we saturate the magnetization by an external magnetic field or if we characterize spin-wave transport using another antenna with no domain wall in its vicinity (Supplementary Fig. 3). The non-uniformities are therefore not caused by the domain wall, but rather by the granular structure of the CoFeB film, as illustrated by optical and atomic-force microscopy images in Supplementary Fig. 4.

**Micromagnetic simulations**. We performed micromagnetic simulations in MuMax3[39] to further analyze spin-wave transmission trough 90° Néel walls. In the simulations, we considered two 20-μm-wide stripe domains. We used two-dimensional periodic boundary conditions and added a 1 μm wide region with higher damping parameter at the edges of the simulation area to prevent spin-wave interference. We ensured that results for this geometry are identical to those of larger area simulations without periodic boundary conditions along $x$ and regions of higher damping. The structure was discretized using finite-difference 2.4 nm × 9.6 nm × 12.5 nm cells. To mimic anisotropy modulations in the experimental system, we abruptly rotated the uniaxial anisotropy axis by 90° at the domain boundary. Input parameters were derived from experiments. We extracted $M_s = 1.15 \times 10^6$ A/m from vibrating sample magnetometry and $K_u = 2.5 \times 10^4$ J/m$^3$ from BLS for our 50-nm-thick CoFeB film on BaTiO$_3$. Additionally, we used an exchange constant of $A_{ex} = 2.1 \times 10^{-11}$ J/m

and a damping parameter of $\alpha = 0.005$. For these parameters, we simulated a domain wall width of 50 nm (head-to-tail wall) and 1.6 μm (head-to-head and tail-to-tail wall). Spin waves were excited locally by an out-of-plane sinusoidal magnetic field at the center of one of the domains, i.e., 10 μm from the pinned domain wall. More details on micromagnetic simulations are given in the Methods section.

Figure 4 summarizes simulation results for spin-wave transmission through a 90° head-to-head (a, c) and 90° head-to-tail (d, f) domain wall at a frequency of 10.85 GHz. In the head-to-head configuration, the effective magnetic field reduces gradually inside the domain wall (Fig. 4b) and the magnetization rotates slowly more perpendicular to the wave vector of propagating spin waves (Fig. 1a). Both effects change the spin-wave dispersion relation. While a decrease of effective field shifts the dispersion curve down, magnetization rotation towards the Damon–Eshbach configuration increases its slope and, thereby, the spin-wave group velocity. The larger group velocity within the head-to-head domain wall reduces the decay of propagating spin waves. Consequently, the spin-wave amplitude after passing the domain wall is somewhat larger compared to its value in a single domain. This effect is illustrated by the dashed line in Fig. 4a, showing the envelope function of propagating spin waves if no domain wall is present. Local changes in the dispersion relation also modify the spin-wave wavelength. Tuning of the group velocity and wavelength inside the head-to-head domain wall produces a finite phase shift upon spin-wave transmission, as illustrated by a change of slope in the contour plot of Fig. 4c.

In contrast, spin waves are strongly reflected by the narrow 90° head-to-tail domain wall (Fig. 4d, f). The effective magnetic field inside this domain wall peaks sharply at its center (Fig. 4e). Two field minima at $x = \pm 20$ nm surround this peak. The non-uniform field profile produces a resonance mode that is characterized by two oscillatory out-of-phase antinodes on opposite sides of the domain-wall center (see inset in Fig. 4f and Supplementary movie 1). Propagating spin waves are absorbed and reflected by this domain-wall resonance mode, as illustrated by the undulations of spin-wave maxima and minima in the contour graph of Fig. 4f. The undulations are produced by interference of counter-

propagating spin waves. Because of this interference effect, the spin-wave amplitude in front of the head-to-tail domain wall (Fig. 4d) is reduced in comparison to the head-to-head configuration (Fig. 4a), in agreement with the experimental data of Fig. 3a, d. Similar resonance modes have been simulated previously for 180° domain walls in magnetic nanowires, and it was shown that oscillating antinodes can trigger spin-wave emission[40] or cause resonant reflection of incoming spin waves[22].

Figure 5 shows the frequency dependence of the spin-wave filtering effect. In panel b, we plot the amplitude of spin waves after passing a magnetic domain wall, relative to their amplitude in the same film without domain wall. Obviously, the 90° head-to-tail wall reduces the spin-wave amplitude in the frequency range from 9 to 13 GHz, with almost zero transmission around 11 GHz. This simulation result agrees well with the experimental

data in Fig. 3. Compared to a single domain, a local enhancement of the group velocity inside the broad head-to-head domain wall reduces the attenuation of spin waves. The frequency dependence of the $m_{z,DW}/m_{z,NoDW}$ ratio for this domain wall is qualitatively explained by the variation of $m_z$ with group velocity ($v_g$) and spin-wave scattering time ($\tau$), $m_z(x) \propto \exp(-x/v_g\tau)$, where $v_g\tau$ corresponds to the spin-wave decay length ($l_d$). Since the spin-wave scattering time is approximated by $\tau = 1/2\pi\alpha f$[41], we can write $m_{z,DW}/m_{z,NoDW} \propto \exp(2\pi x \alpha f(1/v_{g,NoDW} - 1/v_{g,DW}))$. Because the spin-wave group velocity is enhanced inside the head-to-head domain wall ($v_{g,DW} > v_{g,NoDW}$), this expression predicts an exponential increase of $m_{z,DW}/m_{z,NoDW}$ with frequency. We emphasize that Fig. 5b depicts the frequency evolution of the $m_{z,DW}/m_{z,NoDW}$ ratio rather than the amplitude of spin waves after passing the head-to-head domain wall ($m_z$,

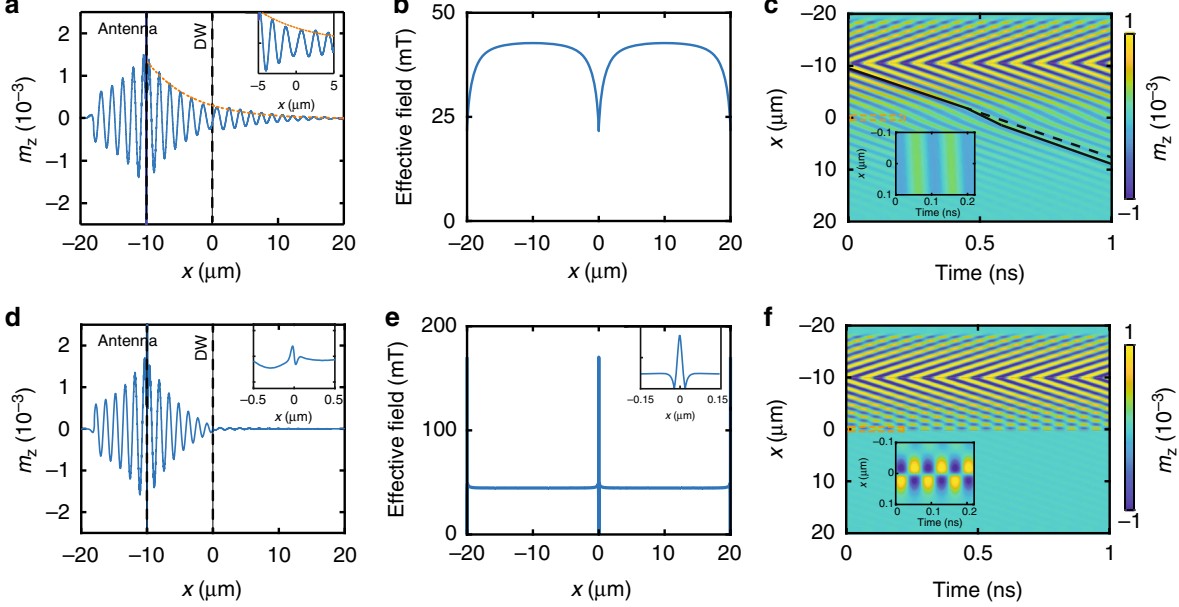

**Fig. 4** Micromagnetic simulations of spin-wave transmission through single domain walls. Spin-wave transport through a broad 90° head-to-head domain wall (**a**, **c**) and a narrow 90° head-to-tail domain wall (**d**, **f**) at a frequency of 10.85 GHz. The panels show data for the steady excitation state, which is reached 9 ns after the sinusoidal magnetic field is turned on. The effective magnetic fields of the two domain walls are plotted in **b** and **e**. The dashed line in **a** depicts the amplitude of propagating spin waves if no domain wall is present. The inset in **f** illustrates strong out-of-phase oscillations of two anti-nodes inside the head-to-tail domain wall

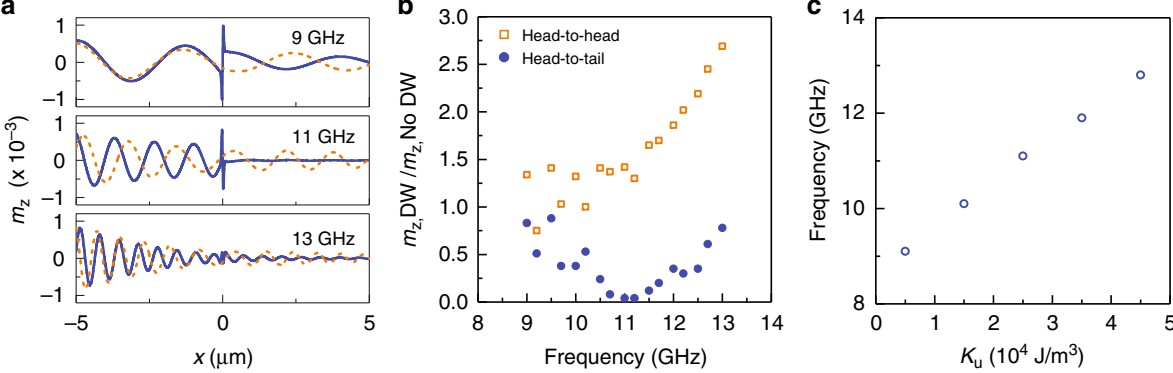

**Fig. 5** Dependence of spin-wave transmission on frequency and anisotropy strength. **a** Micromagnetic simulations of spin-wave transport through a broad 90° head-to-head domain wall (dashed orange line) and a narrow 90° head-to-tail domain wall (solid blue line) at three different frequencies. **b** Frequency dependence of the spin-wave amplitude after passing through a head-to-head and head-to-tail domain wall. The amplitude is normalized to simulation data for a film without a domain wall. **c** Frequency of minimal spin-wave transmission through a head-to-tail domain wall as a function of magnetic anisotropy strength

$_{DW}$). The amplitude of spin waves decreases with frequency because of an overall reduction of the spin-wave decay length, as illustrated by a comparison of the dashed orange curves in Fig. 5a.

**Magnetic spin-wave valve.** Following our results on spin-wave transmission through single 90° domain walls, we propose a new structure for active spin-wave manipulation. Our device concept consists of three stripe domains with uniaxial magnetic anisotropy and two pinned domain walls. In this configuration, magnetization reversal in the central domain switches the domain-wall state between a head-to-head/tail-to-tail combination and two head-to-tail walls (Fig. 1c). In practice, toggling between these two remanent magnetization states can be achieved by applying a magnetic-field pulse along the anisotropy axis of the central domain. Switching broad domain walls into narrow domain walls dramatically changes the transmission of spin waves near the domain-wall resonance frequency. An example at $f = 11$ GHz is shown in Fig. 6. The pinned domain walls are separated by 1.5 μm in this simulation. For the narrow head-to-tail walls, this distance is sufficient to reach a spin rotation of 90° (Fig. 6b). In contrast, the spin rotation is only 38° for the structure with a head-to-head/tail-to-tail wall combination. Irrespective of this finite-size scaling effect[36], the configuration with broad domain walls is fully transparent for propagating spin waves (Fig. 6a, c). Excitation of a resonance mode in the narrow domain walls, on the other hand, reduces spin-wave transmission to nearly 0% (Fig. 6d, f). Since the spin-wave signal can be easily turned on or off by magnetic switching of the central domain, we refer to this structure as a magnetic spin-wave valve[42].

## Discussion
Just like reconfigurable magnonic crystals[11,12,15], properties of pinned 90° domain walls can be changed on demand, enabling their application in magnon conduits, filters, or logic gates. Moreover, since narrow and broad domain-wall states are non-volatile, they can also be used to store data. Downscaling of

programmable domain-wall filters is limited by the head-to-tail domain wall size. For typical materials and uniaxial anisotropy strengths, the width of this domain wall is several tens of nanometers. In nanoscale elements, broad head-to-head/tail-to-tail domain walls cannot form and, consequently, the magnetization of this state is almost uniform[36]. Spin waves propagate through this magnetization configuration without significant pertubation, enabling deterministic switching between a transparent state without domain walls and a strongly reflective state with a narrow head-to-tail wall. To realize switching in patterned devices, one could use local Oersted fields from a metallic nanowire that is placed on top of the ferromagnetic film.

Field control over the width of pinned magnetic domain walls requires an abrupt change of magnetic anisotropy. Here, we used strain coupling in a $CoFeB/BaTiO_3$ bilayer to demonstrate programmable spin-wave filtering. There are, however, other means by which lateral modulations of magnetic anisotropy can be realized. Examples include area-selective ion irradiation[43,44] and thermally assisted scanning probe lithography[45]. Fabrication of pinned domain walls for active spin-wave manipulation is thus not limited to the use of specific substrates.

For the experimental parameters of our system, the transmission of spin waves through head-to-tail domain walls reduces drastically by resonant reflection around 11 GHz. In analogy to magnonic crystals, we estimate a spin-wave bandgap from the transmission curve (Fig. 5b). Using the full width at half maximum, we find a bandgap $\Delta f \approx 2$ GHz. The domain-wall resonance mode and, thereby, the operation frequency of the spin-wave filter can be tuned by varying the anisotropy strength (Fig. 5c). Lowering of the uniaxial magnetic anisotropy reduces the frequency of minimal spin-wave transmission. In our strain-coupled system, this could be achieved by a change of the deposition parameters, the insertion of a seed layer at the ferromagnetic/ferroelectric interface, or the use of a ferromagnetic material with a smaller magnetostriction constant.

In summary, we report on active spin-wave manipulation using programmable domain walls. Control over the transmission of

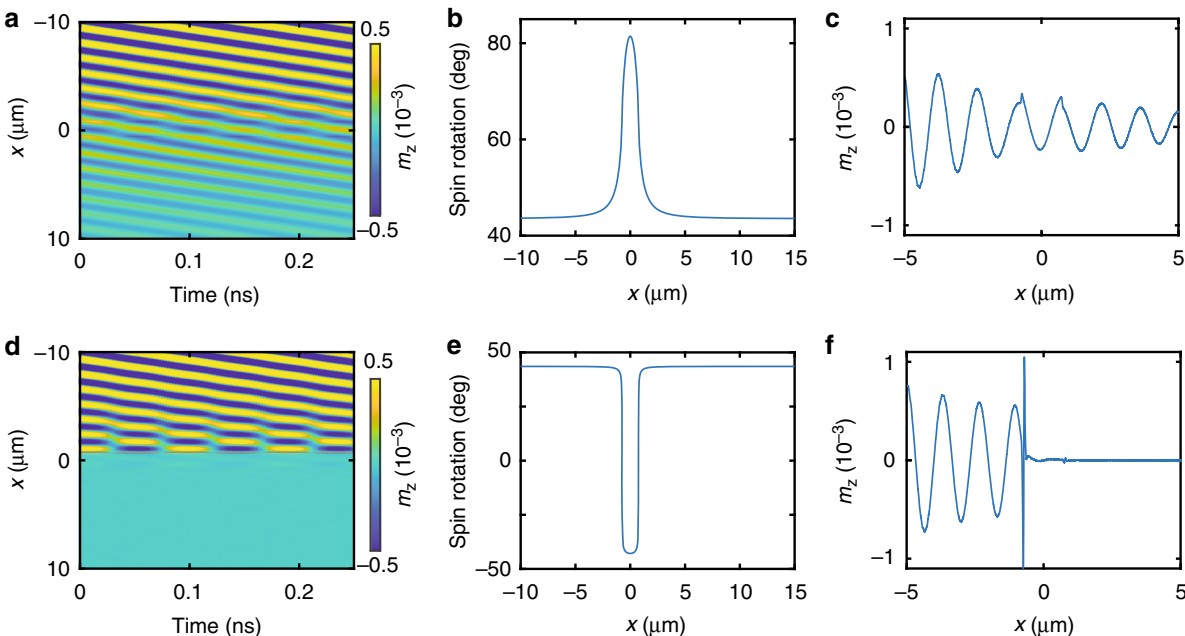

**Fig. 6** Micromagnetic simulations of a magnetic spin-wave valve. In the simulations, the distance between two pinned domain walls is 1.5 μm and the frequency is 11 GHz. Spin waves are excited at $x = -10$ μm, and the panels show data for steady-state excitation. Magnetization reversal in the central domain transforms a head-to-head/tail-to-tail wall combination (transport characteristics in **a**, **c**) into two head-to-tail domain walls (transport results in **d**, **f**), or vice versa. Spin rotations between domains for these two states are shown in **b** and **e**

propagating spin waves is realized by pinning domain walls at 90° anisotropy boundaries. If the domain wall is set to a head-to-head or tail-to-tail configuration, it is broad and transmits propagating spin waves efficiently. Switching to a narrow head-to-tail domain wall reduces the transmission of spin waves to nearly 0% at the domain-wall resonance frequency. Both magnetization configurations are non-volatile and toggling between the two domain-wall states is fully reversible and easily attained by magnetic switching in one of the domains.

## Methods

**Sample preparation**. We grew the 50-nm-thick CoFeB film with a composition of 40% Co, 40% Fe, and 20% B on a single-crystal $BaTiO_3$ substrate using dc magnetron sputtering at 175 °C. At this temperature, $BaTiO_3$ is paraelectric, and its lattice exhibits cubic symmetry. During post-deposition cooling through the paraelectric-to-ferroelectric phase transition at 120 °C, the structure of $BaTiO_3$ becomes tetragonal and a regular pattern of ferroelectric stripe domains forms. The polarization in the domains is oriented in-plane and, together with the elongated axis of the tetragonal unit cell, it rotates by 90° at domain boundaries. The phase transition in the $BaTiO_3$ substrate strains the CoFeB film. Via inverse magnetostriction, this strain induces regular rotations of the uniaxial magnetic anisotropy axis. Since domain walls are strongly pinned by the magnetic anisotropy boundaries, it is possible to control their spin structure by the application of a magnetic field[35]. After cooling to room temperature, we covered the CoFeB film with a 3 nm Ta/28 nm $TaO_x$ bilayer. The $TaO_x$ film was grown by reactive sputtering. Microwave antennas with a width of 500 nm were patterned onto the $TaO_x$ film using electron-beam lithography and lift-off. The broadband antennas consisted of 3 nm Ta and 50 nm Au.

**Magnetic characterization and Brillouin light scattering experiments**. The magnetic domain structure of the sample was imaged using a wide-field magneto-optical Kerr microscope with ×20 and ×100 objectives. We used vibrating sample magnetometry to measure the saturation magnetization of the CoFeB film. From vector network analyzer ferromagnetic resonance (VNA-FMR) measurements, we derived a magnetic damping parameter of 0.007. μ-BLS measurements, with and without phase resolution, were performed by scanning a laser beam spot with a diameter of about 235 nm[46] near the center of an excitation antenna and along the direction of propagating spin waves. The antenna was connected via rf picoprobes to a microwave generator working up to 20 GHz. We selected an antenna with a pinned domain wall in its vicinity. To maintain positional accuracy, we supplied the same reference image as feedback to image recognition-based drift stabilization software in experiments on both head-to-head and head-to-tail domain walls. More details on the μ-BLS setup can be found in ref. [47].

**Micromagnetic simulations**. We performed micromagnetic simulations using open-source GPU-accelerated MuMax3 software. Before spin-wave excitation, the simulation geometry was initialized by aligning the magnetization along the uniaxial anisotropy axes and letting the system reach its ground state in zero magnetic field. The domain-wall widths were calculated for this ground state using the same procedure as in Refs. [35] and[36]. Previous work on high-resolution experimental characterization of pinned 90° magnetic domain walls has shown that micromagnetic simulations provide a good estimate of their width[37,38]. After selection of the domain-wall type, spin waves were excited locally by a 100 mT out-of-plane sinusoidal magnetic field. The ac field was applied to a one-cell-wide line at the center of one of the domains. To visualize propagating spin waves, we recorded the $z$-component of magnetization after reaching steady-state excitation. It took about 9 ns to reach this state. Besides information on propagating spin waves, the simulation signal also contained a dc component caused by ~2° out-of-plane magnetization tilting inside the domain walls. For clarity, we subtracted this dc component from the simulation data.

## Data availability

The data that support the findings of this study are available from the corresponding author upon request.

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

## Acknowledgements

This work was supported by the European Research Council (Grant Nos. ERC-2012-StG 307502-E-CONTROL and ERC-PoC-2018 812841-POWERSPIN) and the Academy of Finland (Grant Nos. 317918 and 316857). S.J.H. acknowledges financial support from the Väisälä Foundation. Lithography was performed at the Micronova Nanofabrication Centre, supported by Aalto University. We also acknowledge the computational resources provided by the Aalto Science-IT project.

## Author contributions

S.J.H. and S.v.D. designed and initiated the research. S.J.H. fabricated the samples. S.J.H., M.M. and G.G. conducted the μ-BLS measurements. S.J.H. and H.J.Q. performed the micromagnetic simulations. S.v.D. supervised the project. S.J.H. and S.v.D. wrote the manuscript, with input from all other authors.

## Additional information

**Competing interests:** The authors declare no competing interests.

