## [Peer Review File · Nature Communications]

Reviewers' comments:

Reviewer #1 (Remarks to the Author):

The manuscript "Programmable control of spin-wave transmission in a domain-wall spin valve" by Sampo J. Hämäläinen et al. presents original measurements and micromagnetic simulations on the spin wave propagation through the 90° Néel domain wall. The authors demonstrate that broad 90° head-to-head domain walls are transparent to spin waves over a broad frequency range but the spin waves are strongly reflected in the narrow 90° head-to-tail domain walls. The usage of CoFeB/BaTiO₃ structure with regular stripe domains as a spin-wave media is novel and the micromagnetic simulation results shows that the so-called "magnetic spin-wave valve" could be used in potential magnonic devices. While this manuscript is well written, I've a few questions below that should be addressed.

1. The authors show lots of simulation results which are very nicely done and helpful to interpret the experimental results and propose novel magnonic devices. However, there is merely one figure (actually only fig2b,c) describing the experimental results in the article. In my opinion, more experiments (e.g. spin wave propagation in different frequencies) should be presented and discussed in order to reach better experiment/simulation balance of the article.

2. What is the size of the single domain structure? It would be better if there are scale bars in fig1c and fig2a.

3. In the BLS measurement, the spin wave amplitude is getting higher when it propagates through the domain wall in fig2c. This is almost incomprehensible because the spin waves leaving the domain wall show larger amplitude than the incident spin waves..... are there any experimental errors?

4. In figure 5, the authors show the prototype of the magnetic spin wave valve where two pinned domain walls structure could be analogue to the spin valve by magnetic simulation. Is it possible for the authors to demonstrate it by experiments? I wonder the high damping of the CoFeB/BaTiO₃ couldn't support the long-distance propagation of spin waves which limit the potential application of the magnonic devices. By BLS measurement, the authors could also investigate the spin wave decay when it propagates through the 90° head-to-tail domain walls. In this case, the authors might want to soften the "100%" argument (well in simulations, it is always possible), since the domain wall itself has finite dimension and we can see that no matter in which magnetic configuration, the spin waves propagating in the sample can only reach maximum 4 microns and then quickly decay and eventually disappear due to relatively large damping.

5. A typo I suppose is on page 4. "... 90 head-to-tail (d-e) domain wall at a frequency of 10.85 GHz". I guess it should refer to "(d-f)"

In general, I find the manuscript well written and the results very interesting. If the authors may address the points raised above, I think this work might be suitable for publication in Nature communications.

Reviewer #2 (Remarks to the Author):

Hämäläinen et al. present their work on the control of spin wave (SW) transmission through two parallel domain walls (DW) in ferromagnetic CoFeB films deposited on the ferroelectric BaTiO₃ bottom layers. The different DW configurations can be toggled by external magnetic fields along or perpendicular to the DWs. Based on the widths of the DWs, wider for head-to-head (tail-to-tail) DWs and narrower for head-to-tail DWs, the authors showed that the transmission of selected frequency of SW can be controlled.

The simulation part of this work is relatively more complete. However, the experimental part is somewhat not convincing and incomplete. In the first part, the authors showed their proposed idea of two parallel DWs and magneto-optical Kerr microscopy images of these DWs in Fig. 1. In the

later part, experimental and simulation results of the SW through one single DW is presented. Finally, more simulation results on one DW, the frequency dependence of the SW transmission and the minimum transmission frequency as a function of the magnetic anisotropy strength is shown in Fig. 4, while results for two pinned DWs are given in Fig. 5.

In Fig. 1 (c), a scale bar is needed. Line profiles perpendicular to the DW is needed to show the DW widths. It should be difficult to resolve a 50 nm DW in optical images as the authors claim.

Since the film thickness and the head-to-tail DW width are both 50 nm, is there any out-of-plane magnetization component observed? BTO usually shows complex FE domain structures. What is the influence of the FE DW width to the CoFeB DW?

What is the spot size of the focused laser beam of the phase-resolved mu-BLS setup? In Fig. 2(c) the authors present two line scans across the domain walls with 200 nm steps. If the intensity can be trusted, why is the point of the black line at 1 micron stronger than the intensity before the SW reaches the DW? The exponential decay of the SW intensity away from the antenna is not observed. For the red line, the intensity between antenna and DW are similar to the black line. How can one reach the conclusion that there is strong reflection of the SW? If there are large error bars, what can we conclude from these measurements? A wide area scan with color coded intensity should be presented before any conclusion can be reached.

In the micromagnetic simulations, are the DW widths, 50 nm and 1.6 micron, results from fully relaxed films with atomically sharp uniaxial anisotropy? Figs. 3 (b) and (e) show the magnitude of the effective fields. What is the direction of the field? Are there any other components in other directions?

Fig. 4 (b), and Fig. 3 (a), shows an amplification of the SW for the head-to-head domain wall. The authors attribute this to the enhanced group velocity. Furthermore, this amplification effect increases with increasing frequency. This sounds unphysical thus deserves more explanation and discussion.

For the SW valve part, only simulation is presented. A more complete experimental data is needed.

For the realization of a SW spin valve, the paper by Wu et al needs to be cited. For the transmission of SW through DWs, two more papers by Buijnsters et al and Chang et al need to be cited, as listed below. Overall, the idea of an in-plane SW valve utilizing a FE material to fulfill the 90-degree DWs is interesting. Unfortunately the switching between different magnetization configurations are by external magnetic fields but not electrical fields. There are some minor questions in simulation results. Experimental results are insufficient and incomplete to support the proposed idea. These results do not fit the high caliber of the Nat. Comm.

The following three papers should be included in the reference.

Buijnsters, F. J., Ferreiros, Y., Fasolino, A. & Katsnelson, M. I. Chirality-dependent transmission of spin waves through domain walls. *Phys. Rev. Lett.* 116, 147204 (2016).

H. Wu, L. Huang, C. Fang, B. S. Yang, C. H. Wan, G. Q. Yu, J. F. Feng, H. X. Wei, and X. F. Han. Magnon Valve Effect between Two Magnetic Insulators. *Phys. Rev. Lett.* 120, 097205 (2018)

Chang, L.-J. et al. Ferromagnetic domain walls as spin wave filters and the interplay between domain walls and spin waves. *Sci. Reports* 8, 3910 (2018).

More papers listed below can be considered.

Allwood, D. A. et al. Magnetic domain-wall logic. *Science* 309, 1688–1692 (2005).

Yan, P., Kamra, A., Cao, Y. & Bauer, G. E. W. Angular and linear momentum of excited ferromagnets. *Phys. Rev. B* 88, 144413 (2013).

Kim, S.-K. Micromagnetic computer simulations of spin waves in nanometre-scale patterned magnetic elements. *J. Phys. D: Appl. Phys* 43, 264004 (2010).

Voto, M., Lopez-Diaz, L. & Torres, L. Effects of grain size and disorder on domain wall propagation in CoFeB thin films. *J. Phys. D: Appl. Phys.* 49, 185001 (2016).

Woo, S., Delaney, T. & Beach, G. S. D. Magnetic domain wall depinning assisted by spin wave bursts. *Nat Phys*

13, 448–454 (2017).

Reviewer #3 (Remarks to the Author):

Dear Editor, Dear Authors,

The manuscript reports on experimental evidence of spin waves transmitted through different types of domain walls and the resulting phase shift. This is one of the holy grails of magnonics and I congratulate the authors for their success. The authors go even further by using the transmission properties of different domain walls for building a non-volatile, zero-field magnon valve.

The paper is scientifically sound, easy to read and from my point of view interesting for the broad readership of Nature Communications because it actually combines research from magnetization dynamics with ferroelectricity, which will open the door for many new ideas.

I recommend the manuscript for rapid publication. The comments below might/could/(and for some editorial remarks should) be addressed but they do not require minor revisions.

Sincerely yours,

Helmut Schultheiß

Comments/Suggestions:

Figure 1c: Please add a scale bar. Furthermore an inset showing the enlarged portion of the domain walls for each case and or a line profile across the walls would nicely show their different width.

Section „Experimental realization“

- the expression „giant reprogramming“ sound strange.

Figure 2a: Scale bar is missing

Regarding the data shown in Fig.2b: For which magnetic configuration was this measured (head to head or head to tail)? Does data for both cases exist? Might be instructive for the read to see the difference, since for the head to tail walls there should be resonances with discrete energies in the domain in stark contrast to a spin wave band in the case of the much wider head to head walls.

Regarding the data shown in Fig.2c: With phase resolved BLS it should in principle be possible to verify the transition from standing waves to propagating waves when changing the domain wall type. Full reconstruction of the phase like shown in Appl. Phys. Lett. 95, 182508 (2009) would give direct evidence. If the data of phase resolved BLS for two different phases of the applied microwave driving signal (90 degree out of phase) and the pure EOM signal and SW signal exists, the reconstruction of the phase profile might be possible and support the micro magnetics even more.

Reviewer #1:

Reviewer comment

The manuscript “Programmable control of spin-wave transmission in a domain-wall spin valve” by Sampo J. Hämäläinen et al. presents original measurements and micromagnetic simulations on the spin wave propagation through the 90° Néel domain wall. The authors demonstrate that broad 90° head-to-head domain walls are transparent to spin waves over a broad frequency range but the spin waves are strongly reflected in the narrow 90° head-to-tail domain walls. The usage of CoFeB/BaTiO₃ structure with regular stripe domains as a spin-wave media is novel and the micromagnetic simulation results shows that the so-called “magnetic spin-wave valve” could be used in potential magnonic devices. While this manuscript is well written, I’ve a few questions below that should be addressed.

Our response

We would like to thank the reviewer for the positive assessment of our work and his/her constructive comments. Below we will address the reviewer’s remarks on a point-by-point basis.

Reviewer comment

1. The authors show lots of simulation results which are very nicely done and helpful to interpret the experimental results and propose novel magnonic devices. However, there is merely one figure (actually only fig2b,c) describing the experimental results in the article. In my opinion, more experiments (e.g. spin wave propagation in different frequencies) should be presented and discussed in order to reach better experiment/simulation balance of the article.

Our response

Stimulated by the reviewer’s comment we have performed additional μ -BLS experiments on our CoFeB/BaTiO₃ sample. New data include local μ -BLS spectra recorded at a fixed position, 2.5 μ m beyond the center of the pinned magnetic domain wall, and by scanning the rf excitation frequency in the range 5 – 15 GHz (new Fig. 2b) and μ -BLS intensity scans across the domain wall at four different frequencies (new Fig. 3 and Supplementary Fig. 2). The new results corroborate that broad 90° head-to-head domain walls are transparent to spin waves over a broad frequency range and that spin waves are strongly reflected by narrow 90° head-to-tail domain walls at the domain-wall resonance frequency. Maximum reflection from the narrow 90° head-to-tail domain wall is measured at an excitation frequency of 10.85 GHz, which agrees well with results from micromagnetic simulations in Fig. 5b (Fig. 4b in the original manuscript).

Reviewer comment

2. What is the size of the single domain structure? It would be better if there are scale bars in fig1c and fig2a.

Our response

The minimal width of the stripe domains in the CoFeB/BaTiO₃ sample is about 16 μ m. We added the requested scale bars in Fig. 1c and Fig. 2a.

Reviewer comment

3. In the BLS measurement, the spin wave amplitude is getting higher when it propagates through the domain wall in fig2c. This is almost incomprehensible because the spin waves leaving the domain wall show larger amplitude than the incident spin waves..... are there any experimental errors?

Our response

We have performed additional phase-resolved μ -BLS measurements to address this issue. The new experiments reveal that an increase of the phase-resolved BLS signal after the head-to-head domain wall occurs at some specific locations along the y-axis, but that its amplitude decays in a more conventional manner at most places. In the revised manuscript, the 2D intensity maps in Fig. 3c,f illustrate the spatial non-uniformity of the BLS response. We attribute this effect to the granular structure of the CoFeB film. To proof this, we included optical- and atomic-force microscopy images of the CoFeB film as Supplementary Fig. 3. Moreover, we have replaced Fig. 2c by a new phase-resolved μ -BLS measurement to avoid confusion. While spatial non-uniformities in the BLS signal are apparent, they do not affect our main conclusion that strong BLS intensities extend beyond broad 90° head-to-head domain walls but not beyond narrow 90° head-to-tail domain walls.

Reviewer comment

4. In figure 5, the authors show the prototype of the magnetic spin wave valve where two pinned domain walls structure could be analogue to the spin valve by magnetic simulation. Is it possible for the authors to demonstrate it by experiments? I wonder the high damping of the CoFeB/BaTiO₃ couldn't support the long-distance propagation of spin waves which limit the potential application of the magnonic devices. By BLS measurement, the authors could also investigate the spin wave decay when it propagates through the 90° head-to-tail domain walls. In this case, the authors might want to soften the "100%" argument (well in simulations, it is always possible), since the domain wall itself has finite dimension and we can see that no matter in which magnetic configuration, the spin waves propagating in the sample can only reach maximum 4 microns and then quickly decay and eventually disappear due to relatively large damping.

Our response

In the experiments, the propagation distance of spin waves is about $6\ \mu\text{m}$ (see BLS curves for the head-to-head domain wall in Figs. 2c and 3a,d). Unfortunately, this is insufficient to pass two pinned domain walls with a minimal separation of $16\ \mu\text{m}$.

In the revised manuscript, we use μ -BLS to experimentally compare spin-wave transmission through broad 90° head-to-head domain walls and narrow 90° head-to-tail domain walls in Figs. 2b,c and 3 as well as Supplementary Fig. 2.

The argument of "100%" spin-wave transmission through broad 90° head-to-head domain walls is based on micromagnetic simulations. We quantify the transmission efficiency by comparing the decay of the spin-wave amplitude in a film with and without a broad 90° head-to-head domain wall (Figs. 4a and 5b). We find that the spin-wave amplitude is slightly larger if the domain wall is present. This effect is explained by an enhancement of the group velocity within the head-to-head domain wall (see text on page 6). The broad 90° head-to-head domain wall does therefore not produce any suppression of the spin-wave signal on top of intrinsic Gilbert damping. In our opinion, this justifies the "100%" argument. We explain this more clearly on page 5 of the revised manuscript (red text). Additionally, we added new experimental data on spin-wave transmission through broad 90° head-to-head domain walls. The μ -BLS intensity measurements depicted in Fig. 3 confirm the high transparency of 90° head-to-head domain walls.

Reviewer comment

5. A typo I suppose is on page 4. "... 90 head-to-tail (d-e) domain wall at a frequency of 10.85 GHz". I guess it should refer to "(d-f)"

In general, I find the manuscript well written and the results very interesting. If the authors may address the points raised above, I think this work might be suitable for publication in Nature communications.

Our response

We corrected the typo.

We are pleased that the reviewer finds our results very interesting and hope that the new experimental data and other revisions convince the reviewer to make a positive recommendation on the publication of our work in Nature Communications.

Reviewer #2:

We would like to thank the reviewer for the assessment of our work and his/her constructive comments. Below we will address the reviewer's remarks on a point-by-point basis.

Reviewer comment

Hamalainen et al. present their work on the control of spin wave (SW) transmission through two parallel domain walls (DW) in ferromagnetic CoFeB films deposited on the ferroelectric BaTiO bottom layers. The different DW configurations can be toggled by external magnetic fields along or perpendicular to the DWs. Based on the widths of the DWs, wider for head-to-head (tail-to-tail) DWs and narrower for head-to-tail DWs, the authors showed that the transmission of selected frequency of SW can be controlled.

The simulation part of this work is relatively more complete. However, the experimental part is somewhat not convincing and incomplete. In the first part, the authors showed their proposed idea of two parallel DWs and magneto-optical Kerr microscopy images of these DWs in Fig. 1. In the later part, experimental and simulation results of the SW through one single DW is presented. Finally, more simulation results on one DW, the frequency dependence of the SW transmission and the minimum transmission frequency as a function of the magnetic anisotropy strength is shown in Fig. 4, while results for two pinned DWs are given in Fig. 5.

In Fig. 1 (c), a scale bar is needed. Line profiles perpendicular to the DW is needed to show the DW widths. It should be difficult to resolve a 50 nm DW in optical images as the authors claim. Since the film thickness and the head-to-tail DW width are both 50 nm, is there any out-of-plane magnetization component observed? BTO usually shows complex FE domain structures. What is the influence of the FE DW width to the CoFeB DW?

Our response

The minimal width of the stripe domains in the CoFeB/BaTiO₃ sample is about 16 μm. We added a scale bar in Fig. 1c to illustrate this.

The domain-wall profiles cannot be resolved by magneto-optical Kerr microscopy. Previously, we have measured the width of pinned magnetic domain walls in CoFeB/BaTiO₃ and related systems using scanning electron microscopy with polarization analysis (SEMPA) and X-ray photoemission electron microscopy (XPEEM) (Refs. 37 and 38 in the revised manuscript). We found that the experimentally measured domain wall widths agree with estimations from micromagnetic simulations. Here, we use this notion to simulate the domain wall widths using experimentally determined input parameters. To clarify this, we added the following text to the micromagnetic simulations paragraph of the Methods section: "Before spin-wave excitation, the simulation geometry was initialized by aligning the magnetization along the uniaxial anisotropy axes and letting the system reach its ground state in zero magnetic field. The domain-wall widths were calculated for this ground state using the same procedure as in Refs. 35 and 36. Previous work on high-resolution experimental characterization of pinned 90° magnetic domain walls has shown that micromagnetic simulations provide a good estimate of their width [37,38]."

In the domain wall, the magnetization tilts out of the plane by about 2°. We subtracted this dc component from the presented m_z data so that the curves depict the amplitude of propagating spin waves. For clarity, we added this information to the micromagnetic simulation paragraph of the Methods section. It now reads: "Besides information on propagating spin waves, the simulation signal also contained a dc component caused by 2° out-of-plane magnetization tilting inside the domain walls. For clarity, we subtracted this dc component from the simulation data."

The BaTiO₃ substrate in our experiments exhibits regular stripe domain with in-plane polarization. The 90° ferroelectric domain walls in BaTiO₃ are only a few nanometers wide. Via strain transfer and inverse magnetostriction these ferroelectric domain walls produce a strong pinning potential for magnetic domain walls in the CoFeB film. The width of pinned magnetic domain walls does not depend on the nearly abrupt ferroelectric domain wall. Instead, it is determined by magnetization alignment (head-to-head/tail-to-tail and head-to-tail configurations produce a broad and narrow domain wall, respectively) and other magnetic parameters (exchange constant, saturation magnetization, magnetic anisotropy). Variation of the magnetic domain wall width upon magnetic switching or a change of the magnetic parameters is discussed in our previous works, most notably in Phys. Rev. B 85, 094423 (2012) and Phys. Rev. Lett. 112, 017201 (2014). References to both papers are included in the manuscript.

Reviewer comment

What is the spot size of the focused laser beam of the phase-resolved μ -BLS setup? In Fig. 2(c) the authors present two line scans across the domain walls with 200 nm steps. If the intensity can be trusted, why is the point of the black line at 1 micron stronger than the intensity before the SW reaches the DW? The exponential decay of the SW intensity away from the antenna is not observed. For the red line, the intensity between antenna and DW are similar to the black line. How can one reach the conclusion that there is strong reflection of the SW? If there are large error bars, what can we conclude from these measurements? A wide area scan with color coded intensity should be presented before any conclusion can be reached.

Our response

Stimulated by the reviewers' questions and comments, we have performed additional μ -BLS measurements on our CoFeB/BaTiO₃ sample. In the revised version of the manuscript, we now present the following experimental data:

1. Local μ -BLS spectra recorded at a fixed position, 2.5 μm beyond the center of the pinned magnetic domain wall, and by scanning the rf excitation frequency in the range 5 – 15 GHz (new Fig. 2b). These new measurements clearly demonstrate that the intensity of propagating spin waves at this location is suppressed strongly when the magnetization configuration is switched from a broad 90° head-to-head domain wall to a narrow 90° head-to-tail domain wall.
2. New phase-resolved μ -BLS scans across the pinned magnetic domain wall (new Fig. 2c). In these measurements, the phase-resolved intensity decays slowly for the broad 90° head-to-head domain wall. In contrast, the narrow 90° head-to-head domain wall suppresses the BLS signal abruptly. An increase of the intensity after the head-to-head domain wall, as observed in the originally submitted graph, is absent now. The granular structure of the CoFeB film caused this anomalous feature (see discussion below).
3. Line scans and requested 2D maps of the BLS intensity for different excitation frequencies (new Fig. 3 and Supplementary Fig. 2). Again, these data reveal that broad 90° head-to-head domain walls are much more transparent to spin waves than narrow 90° head-to-tail domain walls.

The new measurements confirm that reflection of propagating spin waves by head-to-tail walls depends on frequency. The BLS intensity after the pinned head-to-tail domain wall is almost completely suppressed to the noise level at 10.85 GHz (Fig. 3d,e)), whereas a small signal is measured at 9.85 GHz (Fig. 3a,b). In addition, we note that destructive interference of forward moving and reflected spin waves leads to a reduction of the BLS intensity before the head-to-tail domain wall. This suppression is most pronounced at 10.85 GHz, providing additional proof that spin-wave reflection is particularly strong at this frequency. The experimentally observed dependence of spin-wave reflection on frequency agrees with the simulation results of Fig. 5b.

The 2D intensity maps shown in Fig. 3b,c,e,f show spatial non-uniformities in the BLS response. We attribute this effect to the granular structure of the CoFeB film. To proof this, we included optical- and atomic-force microscopy images of the CoFeB film as Supplementary Fig. 3. While spatial non-uniformities in the BLS signal are apparent, they do not affect our main conclusion that strong BLS intensities extend beyond broad 90° head-to-head domain walls but not beyond narrow 90° head-to-tail domain walls.

The spot size of the focused laser beam in the μ -BLS setup was about 235 nm, as demonstrated in G. Gubbiotti, G. Carlotti, M. Madami, S. Tacchi, P. Vavassori and G. Socino, *Setup of a new Brillouin light scattering apparatus with submicrometric lateral resolution and its application to the study of spin modes in nanomagnets*, J. Appl. Phys. 105, 07D521 (2009). We included this information in the BLS paragraph of the Methods section.

Reviewer comment

In the micromagnetic simulations, are the DW widths, 50 nm and 1.6 micron, results from fully relaxed films with atomically sharp uniaxial anisotropy? Figs. 3 (b) and (e) show the magnitude of the effective fields.

What is the direction of the field? Are there any other components in other directions?

Fig. 4 (b), and Fig. 3 (a), shows an amplification of the SW for the head-to-head domain wall. The authors attribute this to the enhanced group velocity. Furthermore, this amplification effect increases with increasing frequency. This sounds unphysical thus deserves more explanation and discussion.

Our response

In our micromagnetic simulations, we consider the magnetic anisotropy boundary to be abrupt. This assumption is justified because the narrow ferroelectric domain wall determines the width of the anisotropy boundary. The magnetic anisotropy boundary is more than one order and two orders of magnitude narrower than the head-to-tail and head-to-head magnetic domain walls, respectively. Information on the simulation geometry and extraction of the magnetic domain wall width are provided on page 5 and page 8 of the revised manuscript.

The simulations in Fig. 3b,e (Fig. 4b,e in the revised manuscript) were performed by aligning the magnetization along the uniaxial anisotropy axes and letting the system reach its ground state in zero magnetic field. In this relaxed state, the effective magnetic field is directed along the magnetization.

In Fig. 3a and Fig. 4b (Fig. 4a and Fig. 5b in the revised manuscript) we compare the decay of the spin-wave amplitude in a film with and without a broad 90° head-to-head domain wall. We find that the spin-wave amplitude is slightly larger if the domain wall is present. We explain this effect as follows: For a single domain (no domain wall), the magnetization and wave vector are oriented at a constant angle of 45°. The spin-wave dispersion relation and group velocity are therefore constant. In the head-to-head domain wall, the magnetization rotates perpendicular to the wave vector (see illustration in the top panel of Fig. 1a). As a result, the slope of the dispersion relation (i.e., group velocity) is locally enhanced compared to that of a single domain without domain wall. Because of this enhancement, the spin-wave amplitude decays more slowly in the broad head-to-head wall and slightly larger amplitudes are simulated behind the wall (Fig. 4a). In the manuscript, we revised the text on page 5 to describe this effect. It now reads: "In the head-to-head configuration, the effective magnetic field reduces gradually inside the domain wall (Fig. 4b) and the magnetization rotates slowly more perpendicular to the wave vector of the propagating spin waves (Fig. 1a). Both effects change the spin-wave dispersion relation. While a decrease of effective field shifts the dispersion curve down, magnetization rotation towards the Damon-Eshbach configuration increases its slope and, thereby, the spin-wave group velocity. The larger group velocity within the head-to-head domain wall reduces the decay of propagating spin waves. Consequently, the spin-wave amplitude after passing the

domain wall is somewhat larger compared to its value in a single-domain without domain wall. This effect is illustrated by the dashed line in Fig. 4a, showing the envelope function of propagating spin waves if no domain wall is present. From this result we conclude that the broad 90° head-to-head domain wall does not produce any suppression of the spin-wave signal on top of intrinsic Gilbert damping (i.e., it is fully transparent to spin waves).”

We now turn to a discussion on the simulation data in Fig. 5b. Here, we plot the amplitude of spin waves after the pinned magnetic domain wall (m_{z_DW}) relative to the amplitude of spin waves at the same location and frequency if no domain wall is present (m_{z_NoDW}). The data illustrate that the ratio between these two amplitudes increases with frequency for the head-to-head domain wall. In the manuscript text, we derive an expression for (m_{z_DW}/m_{z_NoDW}) that supports this observation. The results are physically correct. We emphasize that we depict the m_{z_DW}/m_{z_NoDW} ratio and not the amplitude of spin waves after the pinned domain wall (m_{z_DW}) in Fig. 5b. The amplitude of spin waves decreases with increasing frequency (see dashed orange curves in Fig. 5a) because of an overall reduction of the spin-wave decay length. Figure 5b thus shows that the decrease of m_{z_DW} with frequency is relatively smaller than the decrease of m_{z_NoDW} . To avoid confusion, we have added the following texts: “Compared to a single domain, a local enhancement of the group velocity inside the broad head-to-head domain wall reduces the attenuation of spin waves.” and “We emphasize that Fig. 5b depicts the frequency evolution of the m_{z_DW}/m_{z_NoDW} ratio rather than the amplitude of spin waves after passing the head-to-head domain wall (m_{z_DW}). The amplitude of spin waves decreases with frequency because of an overall reduction of the spin-wave decay length, as illustrated by a comparison of the dashed orange curves in Fig. 5a.”

Reviewer comment

For the SW valve part, only simulation is presented. A more complete experimental data is needed.

Our response

In the experiments, the propagation distance of spin waves is about 6 μm (see BLS curves for the head-to-head domain wall in Figs. 2c and 3a,d). Unfortunately, this is insufficient to pass two pinned domain walls with a minimal separation of 16 μm . In the revised manuscript, we provide new experimental data on spin-wave transmission through pinned 90° head-to-head and 90° head-to-tail domain walls (Figs. 2b,c, Fig. 3 and Supplementary Fig. 2). The new results corroborate that broad 90° head-to-head domain walls are transparent to spin waves over a broad frequency range and that spin waves are strongly reflected by narrow 90° head-to-tail domain walls. We measure nearly complete reflection of propagating spin waves by narrow 90° head-to-tail domain walls at an excitation frequency of 10.85 GHz (Fig. 3). These experimental findings are in good agreement with the micromagnetic simulations of Figs. 4 and 5. Based on the results of spin-wave filtering in pinned 90° magnetic domain walls, we propose a lateral spin-wave valve structure that comprises two parallel domain walls. In the spin-wave valve, reversible magnetic switching in the central domain changes the domain-wall structure from two broad walls to two narrow walls, or vice versa, and, thereby, the transmission of spin waves turns off or on (micromagnetic simulations in Fig. 6). The ability to control propagating spin waves by magnetic switching of a small magnetic domain (the minimal size of the domain is set by the width of the narrow head-to-tail domain wall) is advantageous for device integration and its use as logic or non-volatile memory element. We emphasize that the lateral spin-wave valve is presented as a new concept (supported by experiments and simulations on single domain walls) and that the required modulation of magnetic anisotropy could be realized by other methods such as ion-beam implantation.

Reviewer comment

For the realization of a SW spin valve, the paper by Wu et al needs to be cited. For the transmission of SW through DWs, two more papers by Buijnsters et al and Chang et al need to be cited, as listed below. Overall, the idea of an in-plane SW valve utilizing a FE material to fulfill the 90-degree DWs is interesting. Unfortunately the switching between different magnetization configurations are by external magnetic fields but not electrical fields. There are some minor questions in simulation results. Experimental results are insufficient and incomplete to support the proposed idea. These results do not fit the high caliber of the Nat. Comm.

The following three papers should be included in the reference.

Buijnsters, F. J., Ferreiros, Y., Fasolino, A. & Katsnelson, M. I. Chirality-dependent transmission of spin waves through domain walls. *Phys. Rev. Lett.* 116, 147204 (2016).

H. Wu, L. Huang, C. Fang, B. S. Yang, C. H. Wan, G. Q. Yu, J. F. Feng, H. X. Wei, and X. F. Han. Magnon Valve Effect between Two Magnetic Insulators. *Phys. Rev. Lett.* 120, 097205 (2018).

Chang, L.-J. et al. Ferromagnetic domain walls as spin wave filters and the interplay between domain walls and spin waves. *Sci. Reports* 8, 3910 (2018).

More papers listed below can be considered.

Allwood, D. A. et al. Magnetic domain-wall logic. *Science* 309, 1688–1692 (2005).

Yan, P., Kamra, A., Cao, Y. & Bauer, G. E. W. Angular and linear momentum of excited ferromagnets. *Phys. Rev. B* 88, 144413 (2013).

Kim, S.-K. Micromagnetic computer simulations of spin waves in nanometre-scale patterned magnetic elements. *J. Phys. D: Appl. Phys* 43, 264004 (2010).

Voto, M., Lopez-Diaz, L. & Torres, L. Effects of grain size and disorder on domain wall propagation in CoFeB thin films. *J. Phys. D: Appl. Phys.* 49, 185001 (2016).

Woo, S., Delaney, T. & Beach, G. S. D. Magnetic domain wall depinning assisted by spin wave bursts. *Nat Phys.* 13, 448–454 (2017).

Our response

We added the papers by Wu et al., Buijnsters et al., Chang et al., and Woo et al. to the manuscript. In the revised version of our manuscript, we substantially increased the amount of experimental evidence in support of the main conclusion that switching of a pinned 90° magnetic domain wall from a broad head-to-head to a narrow head-to-tail configuration can be used to actively control the transmission of propagating spin waves. We hope that our new experimental data and other revisions convince the reviewer to make a positive recommendation on the publication of our work in Nature Communications.

Reviewer #3:

Reviewer comment

The manuscript reports on experimental evidence of spin waves transmitted through different types of domain walls and the resulting phase shift. This is one of the holy grails of magnonics and I congratulate the authors for their success. The authors go even further by using the transmission properties of different domain walls for building a non-volatile, zero-field magnon valve. The paper is scientifically sound, easy to read and from my point of view interesting for the broad readership of Nature Communications because it actually combines research from magnetization dynamics with ferroelectricity, which will open the door for many new ideas. I recommend the manuscript for rapid publication. The comments below might/could/(and for some editorial remarks should) be addressed but they do not require minor revisions.

Our response

We would like to thank the reviewer for the positive assessment of our work and his/her recommendation for rapid publication in Nature Communications. Below we will address the reviewer's comments on a point-by-point basis.

Reviewer comment

Figure 1c: Please add a scale bar. Furthermore a inset showing the enlarged portion of the domain walls for each case and or a line profile across the walls would nicely show their different width.

Our response

We added a scale bar to Fig. 1(c). The images in this figure are recorded by magneto-optical Kerr microscopy. The spatial resolution of this optical technique is insufficient to resolve the line profiles of the domain walls. In previous studies, we measured the profiles of head-to-head and head-to-tail domain walls in CoFeB/BaTiO₃ and related systems using scanning electron microscopy with polarization analysis (SEMPA) and X-ray photoemission electron microscopy (XPEEM). References to these works are provided in the manuscript (Refs. 37 and 38).

Reviewer comment

Section „Experimental realization“
- the expression „giant reprogramming“ sound strange.

Our response

We changed the text to “... with the ability to reprogram the domain wall width ...”.

Reviewer comment

Figure 2a: Scale bar is missing.

Our response

We add a scale bar to Fig. 2(a).

Reviewer comment

Regarding the data shown in Fig.2b: For which magnetic configuration was this measured (head to head or head to tail)? Does data for both cases exist? Might be instructive for the read to see the difference, since for the head to tail walls there should be resonances with discrete energies in the domain in stark contrast to a spin wave band in the case of the much wider head to head walls.

Our response

We replaced the data in Fig. 2(b) with local μ -BLS frequency scans for both domain-wall configurations. The new data nicely demonstrate that the head-to-head domain wall is much more transparent to spin waves than the head-to-tail domain wall.

Reviewer comment

Regarding the data shown in Fig.2c: With phase resolved BLS it should in principle be possible to verify the transition from standing waves to propagating waves when changing the domain wall type. Full reconstruction of the phase like shown in Appl. Phys. Lett. 95, 182508 (2009) would give direct evidence. If the data of phase resolved BLS for two different phases of the applied microwave driving signal (90 degree out of phase) and the pure EOM signal and SW signal exists, the reconstruction of the phase profile might be possible and support the micro magnetics even more.

Our response

We thank the reviewer for this suggestion. Unfortunately, due to non-uniformities caused by the granular structure of the CoFeB film and interference between propagating spin waves and spatially uniform magnetization precessions near the antenna, it is impossible to perform the proposed analysis. In the revised manuscript, we present additional experimental data on the frequency dependence of the BLS intensity for both domain-wall configurations (new Fig. 3 and Supplementary Fig. 2). These new data indicate maximum reflection of spin waves by the head-to-tail domain wall at a frequency of 10.85 GHz, which agrees nicely with the simulation results of Fig. 5(b).

Reviewers' comments:

Reviewer #1 (Remarks to the Author):

In their revised version of manuscript and Supplementary material, the authors did take into account all the comments made by reviewers in the first round. The authors put additional μ -BLS data and the balance between the experiment and simulation in the main text and in the Supplementary material have been improved. The reference list has also been improved. The explanation of the broad 90° head-to-head domain walls transparent to spin waves than narrow 90° head-to-tail domain walls is more convincing.

However, there are still some points that should be addressed. The authors claim programmable control of spin wave transmission in domain wall spin valve in title. However they mainly demonstrated programmable spin-wave filtering effect by domain walls. The Fig. 6 of domain wall spin valve micromagnetic simulation is not sufficient to support the title. Of course it is fine to put the claim of domain wall spin valve in the abstract, but not in the title.

In Fig 3c and f, there is spatial non-uniformities in the measured spin-wave intensity. The authors attributed this to the granular structure of the CoFeB film, which is illustrated by optical and AFM images. To me the non-uniformities is so much that it is not so simply caused by the granular structure. For example, is it possible that the broad 90° head-to-head domain wall is not perfect that spin textures of some parts are better formed that more transparency is induced? If this is the case, It might be possible that even if the CoFeB film is relatively uniform, the spatial spin-wave intensity is different.

In Fig. 4, the authors claim head-to-head domain wall does not produce any suppression of the spin-wave signal on top of intrinsic Gilbert damping, which means fully transparent to spin waves. In actual experiment of this manuscript, spin textures of CoFeB are formed by the strain transfer from the substrate. The samples of this kind always results in a larger Gilbert damping compared with intrinsic damping of a well grown CoFeB film without strain effect from the substrate. The assumption of intrinsic damping in sample of this kind is almost unfeasible. So the claim head-to-head domain wall does not produce any suppression of the spin-wave signal on top of intrinsic Gilbert damping (i.e., it is fully transparent to spin waves) should be tuned down.

The manuscript may be suitable for publication in Nature Communications after addressing the previous mentioned issues properly.

Reviewer #2 (Remarks to the Author):

Hamalainen et al. present very convincing experimental BLS data of spin wave transmission through different domain wall configurations in this new version. Since one reprogrammable domain wall can fulfill the control of spin-wave transmission, the spin valve composed of two domain walls proposed by the authors seems redundant. It is obvious that multiple domain walls works better than a single one. Does the proposed spin valve works better than the sum of two individual domain walls? Does the resonance frequency of the domain walls change in the spin valve structure? If both answers are no, the proposed spin valve idea can be replaced by the following statement. 'In the CoFeB/BaTiO₃ bilayer, multiple domain walls can be easily reprogrammed to control the spin wave transmission.'

I would suggest the authors to make the following, not mandatory, changes.

1. Change the title to: Control of spin-wave transmission by a programmable domain wall.
2. Change the subtitle 'Magnetic spin-wave valve' to 'Effect of multiple domain walls'.

This manuscript can be accepted for publication.

Reviewer #3 (Remarks to the Author):

I am happy with the revised manuscript. I also believe that the additional experimental data on 2D mapping of BLS intensities and the different domain wall types is improving the paper a lot. The authors addressed all my comments to my full satisfaction.

Reviewer #1:

Reviewer comment

In their revised version of manuscript and Supplementary material, the authors did take into account all the comments made by reviewers in the first round. The authors put additional μ -BLS data and the balance between the experiment and simulation in the main text and in the Supplementary material have been improved. The reference list has also been improved. The explanation of the broad 90° head-to-head domain walls transparent to spin waves than narrow 90° head-to-tail domain walls is more convincing.

However, there are still some points that should be addressed. The authors claim programmable control of spin wave transmission in domain wall spin valve in title. However they mainly demonstrated programmable spin-wave filtering effect by domain walls. The Fig. 6 of domain wall spin valve micromagnetic simulation is not sufficient to support the title. Of course it is fine to put the claim of domain wall spin valve in the abstract, but not in the title.

Our response

We would like to thank the reviewer for his/her second assessment and constructive comments. The reviewer rightly notes that the manuscript mainly focuses on programmable spin-wave filtering by domain walls. Therefore, we changed the title to: "Control of spin-wave transmission by a programmable domain wall".

Reviewer comment

In Fig 3c and f, there is spatial non-uniformities in the measured spin-wave intensity. The authors attributed this to the granular structure of the CoFeB film, which is illustrated by optical and AFM images. To me the non-uniformities is so much that it is not so simply caused by the granular structure. For example, is it possible that the broad 90° head-to-head domain wall is not perfect that spin textures of some parts are better formed that more transparency is induced? If this is the case, It might be possible that even if the CoFeB film is relatively uniform, the spatial spin-wave intensity is different.

Our response

We tested the reviewer's assertion that non-uniformities in the BLS intensity maps of Fig. 3c and f could be due to imperfections in the spin texture of the head-to-head domain wall by performing additional BLS measurements. In the new experiments, we measured the BLS intensity on the same structure in a saturating bias field (i.e. uniform magnetization, no domain wall) and in zero field on another structure where no domain wall is present near the excitation antenna. In both cases, non-uniformities in the BLS intensity, similar to those in Fig. 3c and f, are recorded. From these results, we conclude that the granular structure of the CoFeB film is mainly responsible for the non-uniformities.

We added the new BLS intensity maps to the Supplementary Information (new Supplementary Fig. 3) and on page 4 of the manuscript we now write: "In the 2D scans, we observe spatial non-uniformities in the spin-wave intensity. Similar BLS-intensity variations are also measured if we saturate the magnetization by an external magnetic field or if we characterize spin-wave transport using another antenna with no domain wall in its vicinity (Supplementary Fig. 3). The non-uniformities are therefore not caused by the domain wall, but rather by the granular structure of the CoFeB film, as illustrated by optical and atomic-force microscopy images in Supplementary Fig. 4."

Reviewer comment

In Fig. 4, the authors claim head-to-head domain wall does not produce any suppression of the spin-wave signal on top of intrinsic Gilbert damping, which means fully transparent to spin waves. In actual experiment of this manuscript, spin textures of CoFeB are formed by the strain transfer from the substrate. The samples of this kind always results in a larger Gilbert damping compared with intrinsic damping of a well grown CoFeB film without strain effect from the substrate. The assumption of intrinsic damping in sample of this kind is almost unfeasible. So the claim head-to-head domain wall does not produce any suppression of the spin-wave signal on top of intrinsic Gilbert damping (i.e., it is fully transparent to spin waves) should be tuned down.. What is the size of the single domain structure? It would be better if there are scale bars in fig1c and fig2a.

Our response

We wanted to emphasize that a head-to-head domain wall does not suppress spin waves more than a single domain without domain wall in the same film. We understand that the use of “intrinsic Gilbert damping” is confusing and incorrect. We therefore removed the sentence from the manuscript. Scale bars are shown in Fig. 1c and 2a.

Reviewer #2:

Reviewer comment

Hamalainen et al. present very convincing experimental BLS data of spin wave transmission through different domain wall configurations in this new version. Since one reprogrammable domain wall can fulfill the control of spin-wave transmission, the spin valve composed of two domain walls proposed by the authors seems redundant. It is obvious that multiple domain walls works better than a single one. Does the proposed spin valve works better than the sum of two individual domain walls? Does the resonance frequency of the domain walls change in the spin valve structure? If both answers are no, the proposed spin valve idea can be replaced by the following statement. 'In the CoFeB/BaTiO₃ bilayer, multiple domain walls can be easily reprogrammed to control the spin wave transmission.'

I would suggest the authors to make the following, not mandatory, changes.

1. Change the title to: Control of spin-wave transmission by a programmable domain wall.
2. Change the subtitle 'Magnetic spin-wave valve' to 'Effect of multiple domain walls'.

This manuscript can be accepted for publication.

Our response

We would like to thank the reviewer for his/her second assessment and constructive comments. We are happy that the reviewer finds our new BLS data of spin wave transmission through different domain wall configurations very convincing and recommends publication of our work. Following his/her suggestion, we changed the title of the manuscript to: "Control of spin-wave transmission by a programmable domain wall".

Reviewer #3:

Reviewer comment

I am happy with the revised manuscript. I also believe that the additional experimental data on 2D mapping of BLS intensities and the different domain wall types is improving the paper a lot. The authors addressed all my comments to my full satisfaction.

Our response

We would like to thank the reviewer for the positive assessment of our work.

REVIEWERS' COMMENTS:

Reviewer #1 (Remarks to the Author):

In the revised version of manuscript and Supplementary material, the authors did take into account all the comments made by reviewers. The authors made adjustment to the title of the manuscript properly. It is also clarified that the non-uniformities in the BLS intensity is not due to the imperfections of the spin texture of the head-to-head domain wall but the granular structure of the film. The BLS experimental results of the spin wave transmission through the control of different type of domain walls are now very convincing to me.

Nevertheless, I have two final questions or suggestions (not mandatory):

1. Is it possible that the authors give an estimate of the damping parameter of the film experimentally, such as by FMR measurement? People who want to follow their work may wonder about the damping parameter for further spin wave manipulation and functionality.

2. One more concern is about the simulation. The authors used two-dimensional periodic boundary conditions and added a 1 μm wide region with higher damping parameter at the edge of the simulation area to prevent spin-wave interference. Could the author please explain how they used the periodic boundary condition (in which two dimensions) and where they add the high damping region? My concern is that the high damping region of 1 μm is comparable to the simulating region of 20 μm . Does the dipolar effect of this high damping region influence the simulation results? The results are probably not too different but I am curious about how much difference it can make.

In principle, I am in favor of publishing this work in Nature Communications.

Reviewer #1:

Reviewer comment

In the revised version of manuscript and Supplementary material, the authors did take into account all the comments made by reviewers. The authors made adjustment to the title of the manuscript properly. It is also clarified that the non-uniformities in the BLS intensity is not due to the imperfections of the spin texture of the head-to-head domain wall but the granular structure of the film. The BLS experimental results of the spin wave transmission through the control of different type of domain walls are now very convincing to me.

Nevertheless, I have two final questions or suggestions (not mandatory):

1. Is it possible that the authors give an estimate of the damping parameter of the film experimentally, such as by FMR measurement? People who want to follow their work may wonder about the damping parameter for further spin wave manipulation and functionality.

Our response

We measured the damping parameter of the film using VNA-FMR and added the result, 0.007, to the Methods section.

Reviewer comment

2. One more concern is about the simulation. The authors used two-dimensional periodic boundary conditions and added a 1 μm wide region with higher damping parameter at the edge of the simulation area to prevent spin-wave interference. Could the author please explain how they used the periodic boundary condition (in which two dimensions) and where they add the high damping region? My concern is that the high damping region of 1 μm is comparable to the simulating region of 20 μm . Does the dipolar effect of this high damping region influence the simulation results? The results are probably not too different but I am curious about how much difference it can make.

Our response

In our simulations, we used an area of 40 μm (two domains of 20 μm) along x and considered a 1 μm wide region of higher damping at the edges (i.e. at $x = -20 \mu\text{m}$ and $x = +20 \mu\text{m}$). This configuration was chosen to limit spin-wave interference and simulation time. To ensure that our simulation geometry does not introduce artefacts, we performed test simulations on larger areas without periodic boundary conditions along x and without regions of higher damping (see figure below). The difference in the spin-wave profile is negligible at and beyond the pinned domain wall.

Comparison of a simulation performed on a 80 μm wide area without periodic boundary conditions along x (NO PBC) and without regions of higher damping at the edges and a simulation from the manuscript (PBC). In these simulations, a head-to-head domain wall is pinned at $x = 0 \mu\text{m}$.